# The Role of Mitochondria in Immune-Cell-Mediated Tissue Regeneration and Ageing

**DOI:** 10.3390/ijms22052668

**Published:** 2021-03-06

**Authors:** Yu-Jih Su, Pei-Wen Wang, Shao-Wen Weng

**Affiliations:** 1Department of Internal Medicine, Kaohsiung Chang Gung Memorial Hospital, Kaohsiung 833, Taiwan; 2College of Medicine, Chang Gung University, Kaohsiung 833, Taiwan; 3Center for Mitochondrial Research and Medicine, Kaohsiung Chang Gung Memorial Hospital, 123, Dapi Road, Niaosong District, Kaohsiung 833, Taiwan; bensu8@gmail.com (Y.-J.S.); wangpw@adm.cgmh.org.tw (P.-W.W.)

**Keywords:** mitochondria, regeneration, inflammation, ageing

## Abstract

During tissue injury events, the innate immune system responds immediately to alarms sent from the injured cells, and the adaptive immune system subsequently joins in the inflammatory reaction. The control mechanism of each immune reaction relies on the orchestration of different types of T cells and the activators, antigen-presenting cells, co-stimulatory molecules, and cytokines. Mitochondria are an intracellular signaling organelle and energy plant, which supply the energy requirement of the immune system and maintain the system activation with the production of reactive oxygen species (ROS). Extracellular mitochondria can elicit regenerative effects or serve as an activator of the immune cells to eliminate the damaged cells. Recent clarification of the cytosolic escape of mitochondrial DNA triggering innate immunity underscores the pivotal role of mitochondria in inflammation-related diseases. Human mesenchymal stem cells could transfer mitochondria through nanotubular structures to defective mitochondrial DNA cells. In recent years, mitochondrial therapy has shown promise in treating heart ischemic events, Parkinson’s disease, and fulminating hepatitis. Taken together, these results emphasize the emerging role of mitochondria in immune-cell-mediated tissue regeneration and ageing.

## 1. Introduction

Organ injuries require tissue repair and regeneration, while subtle tissue injury also leads to macrophage activation, and fibroblast or stromal cell proliferation, such as plaque formation in atherosclerosis. Three phases are involved in the healing process: modifying the inflammatory process, cell proliferation, and remodeling the extracellular matrix. Tissue inflammation is clearly defined by the presence of either inflammatory cells or cytokines in the tissue. However, the differences between tissue repair, tissue regeneration, and tissue proliferation are subtle. Natural tissue repair occurs after tissue inflammation or damage, and the transition from inflammation to repair occurs automatically and sequentially. Importantly, the switch between wound repair, and the regeneration and inflammation pathways lead to differing disease outcomes [1]. In general, tissue repair and tissue regeneration are different steps of a continuum. Tissue repair is an early stage of functional restoration, but lacks patterned three-dimensional tissue reconstruction, which is expressed in tissue regeneration [2]. However, unhindered tissue regeneration, occurring under unknown mechanisms, leads to tissue proliferation, which could further cause unwanted insults, such as fibrosis in ageing. 

Both innate and adaptive immunity against environmental stimulus needs energy supplies. The interaction between immune activation and the energy supplies is based on immuno-metabolomics. In short, the energy requirements for fast responses, such as the neutrophils, the dendritic cell, or the macrophages, mostly come from available glucose, i.e., glycolytic response, or glycogen storage inside the cell [3]. After the first rapid response finishes, the immune systems require energy supplies for further continuous reactions, such as differentiations, transformation, phagocytosis, or killing extracellular pathogens. At the same time, the energy supply mostly shifts from the glycolysis toward either the tricarboxylic acid (TCA) cycle or fatty acid beta-oxidation [3,4].

The mitochondrion has an indispensable role in the development and function of immune cells, from energy supply to the activation, proliferation, and phagocytosis, and ending with repairs and regeneration. Besides energy supply, with the help of mitochondrial reactive oxygen species (ROS), the immune cells execute complex functions, such as apoptosis, pyroptosis, netosis, and adaptive immunity activation. The mitochondrial ROS helps T cell activation and B cell antibody production [3,5].

In this review article, we focus on the role of immune cells in various tissue stages, from inflammation, through repair and regeneration, to tissue proliferation, with an emphasis on the emerging role of mitochondria in immune-cell-mediated tissue regeneration and ageing.

## 2. Immune Cells in Tissues under a Homeostasis Condition

In homeostasis, the immune cells’ continued surveillance involves the innate immunity and adaptive immunity systems working collaboratively to maintain balance between environmental stimuli, and prevent unwanted pathogen invasion. Collectively, innate immunity involves the stromal cells forming the skin barrier, endothelial cells forming the vessel barrier, and epithelial cells forming the mucosal barrier, as well as the leukocytes, natural killer cells, dendritic cells, and macrophages. Meanwhile, adaptive immunity including T cells and B cells is on regular immune surveillance, with the plasma B cells producing immunoglobulin to counter invading pathogens, and the memory T cells remaining dormant, awaiting stimulation to trigger activation. The crosstalk between innate immunity and adaptive immunity depends on the antigen-presenting cells, such as dendritic cells or macrophages, presenting constant signals to T cells. If the antigen-presenting cells present self-antigen to T cells, this causes the T cells to become anergic, thereby becoming regulatory T cells (Treg). Meanwhile, the cytokine profiles are more likely to be anti-inflammatory cytokines, such as IL-4, IL-10, or TGF-beta. The immune cells, either innate or adaptive immunity cells, are capable of moving outside the vessels and rolling toward the homing location according to cytokine attraction, or the adhesion molecules’ traction. In general, under the homeostasis condition, all players in the immune system, including Th1, Th2, Th17 cells, myeloid cells, and B cells, interact with normal pathogens without clinical inflammation [6]. 

The size, morphology, and number of mitochondria inside the cell are important clues to immune cell activities and functions. Generally speaking, the enzymes, either from the TCA cycle or from the beta-oxidation, may be a clue to mitochondrial activity elevations in cells. Different subtypes of immune cells favor different energy resources, and the resources of energy can be diverse within each cell type. Take macrophages for example, the M1 cells use not only glycolysis, but also the pentose phosphate pathway, while their TCA cycle is impaired, while M2 and mature T cells are thought to be dependent on β-oxidation associated with an increased TCA, increased respiratory capacity, and increased production of ATP by oxidative phosphorylation [7,8]. Mitochondrial dynamics, and fusion and fission events also affect immune function. Mitochondrial dynamics mostly affect the subsequent adaptive immune reaction rather than the initiation of innate immunity. In 2016, Buck et al. demonstrated that mitochondrial fusion is associated with memory T cell formation, and improves the T cell function against tumors [9]. Besides adaptive immunity, macrophages use mitofusin 2 protein against RNA virus infection [10]. In a cybrid model of diabetes, our group studied the genetic effect of mitochondria in cells with the same nuclear genomic background, and demonstrated the role of mitochondrial dynamics in regulating immunity and mitochondrial ROS [11,12]. These data exemplified that mitochondrial dynamics have a role in linking cellular immunity and energetics.

## 3. Immune Cells under Inflammatory and/or Injury Conditions

During tissue injury, the immediate response immune cells (innate immunity) respond to the alarms sent from injured cells in differing ways; with the adaptive immunity mechanisms subsequently joining the inflammatory reaction. The overall scope of inflammation depends on the extent of injury. The control mechanism of each immune reaction relies on the orchestration of different types of helper T cells and the activators, signal 1, 2, and 3, antigen-presenting cells, co-stimulatory molecules, and cytokines. After helper T cells (Th) are activated, either Th1, Th2, or Th17 operate and summon innate immunity cells to counteract the invasive pathogens [13,14]. The B cells are also activated, and produce relevant antibodies to combat the antigen, and through the mechanisms of isotype switch, and affinity maturation, the binding of the antibody to the antigen becomes more specific with time. To illustrate this, in lupus nephritis, the immune complex deposition on the glomerular of the kidneys can activate innate and adaptive immunity mechanisms to join and clear the battlefield. Either action could lead to organ inflammation and tissue damage, and ensuing proteinuria or hematuria, which is called glomerulonephritis. Before and during clinical evidence of glomerulonephritis, the tissue damage has already started, and necessary tissue repair is underway. The M1 macrophage is a pro-inflammatory cell that secretes pro-inflammatory cytokines, such as interferon-γ (IFN-γ), interleukins (IL) (IL-1, IL-6, IL-12, IL-23), and tumor necrosis factor-α (TNF-α) [15,16,17]; while the M2 macrophage is believed to be an anti-inflammatory phenotype [18], dependent on several other subsets of anti-inflammatory cytokines, including IL-4 and IL-13 [19]. Our heretofore unpublished data demonstrated that the macrophage phenotype had shifted from M1 type to M2 type in some lupus nephritis patients under partial treatment. Although the precise role of M1 macrophages in the inflammation, and M2 macrophages in the repair or regeneration remain unclear, the M1 to M2 switch indicates the initiation of the repair or regeneration in a dynamic condition [1]. 

Tissue regeneration is partially dependent on macrophages [20], regulatory T cells [21], and other cells, but primarily on stem cells [2,22,23,24]. Different organ systems have their own highly specified functions, and the necessary tissue regeneration is thus heavily regulated. Some specified molecules, such as galectins and cytokines, can either positively or negatively regulate inflammation, regeneration, and ageing in different organ systems [25] (Table 1, Figure 1). In heart failure (HF) conditions, elevation of cardiac damage markers (troponins or natriuretic peptides) have been found useful in the diagnosis of HF and in outcome prediction [26]. In the CORONA and COACH trials, repeated measurements of galectin-3 level, a protein involved in myocardial fibrosis and remodeling, provided a important and significant prognostic value for identifying HF [26]. In atherosclerosis plaque, loss of galectin-3 in macrophage has been associated with increased expression of pro-inflammatory genes, including matrix metalloproteinase 12 (MMP-12), chemokine ligand 2 (CCL2), prostaglandin-endoperoxide synthase 2 (PTGS2), and interleukin-6 (IL-6), while the expression of transforming growth factor-β1 (TGFβ1) was reduced, suggesting a prominent protective role of galectin-3 in regulating macrophage polarization and plaque progression [25]. In animal experiments of acute renal failure, galectin-3 expression was markedly up-regulated [27], and galectin-3 from macrophages is considered as a major mechanism in renal fibrosis [22]. Furthermore, galectin-3 has been studied in several models of chronic kidney diseases, including nephropathies induced by ageing, ischemia, diabetes, and chronic allograft injury [24,28]. In skin, galectin-3 facilitates intracellular epidermal growth factor receptor (EGFR) trafficking to the surface, which could promote wound re-epithelialization [23].

## 4. Immunity and the Ageing Condition

Ageing in the immune system occurs in a similar manner as in other organ systems, and several phenotypic and molecular hallmarks of ageing are exhibited in immune cells, including decreased MHC II expression, decreased antibody production, altered toll-like receptor expression, limited diversity in B cell receptor repertoire, and restricted diversity in the T cell receptor repertoire [41]. Recently, the Baltimore Longitudinal Study on Ageing compared CD4+ T cells obtained from younger (20–39 years-old) and older (>70 years-old) healthy participants, demonstrating that mitochondrial respiration was impaired in CD4+ T cells from older subjects. In older age groups, dysfunction of autophagy leads to defective mitochondrial turnover, triggering chronic inflammation and impairment of the immune defense system [42]. While the T cell ageing process is overwhelming, the antigen–antibody responses will be compromised, which leads to so-called immunosenescence. Immunosenescence includes increases in unnecessary terminal-differentiated memory cell populations and altered cytokine production [43]. Senescent cells are characterized by high expressions of cell-cycle inhibitors, including p16, p21, and p53, coupled with a persistent DNA damage response, which can eventually induce a pro-inflammatory state, known as senescence-associated secretory phenotype (SASP), involving interleukin-6 (IL-6), IL-8, monocyte chemoattractant protein-1 (MCP-1), IL-1β, plasminogen activator inhibitor-1 (PAI-1), and high mobility group box 1 (HMGB-1) [44]. In one large cohort study by Piber et al. of the relationship between ageing and inflammation-associated immune biomarkers [45], significant correlations between the age of the blood donor and the serological markers, IL-6 and soluble TNF receptor II were identified. In addition, intracellular signal transducer and activator of transcription (STAT) signaling, which is associated with immunity dysfunction, including STAT1, STAT3, STAT5, was also found to be significantly correlated with age. 

Pinti et al. demonstrated that mitochondrial DNA (mtDNA) plasma levels increased gradually after the fifth decade of life, and there was a role for familiar/genetic background in controlling the levels of circulating mtDNA. The circulated mtDNA was associated with inflammatory cytokines, TNF-α, IL-6, RANTES, and IL-1ra [46]. Furthermore, different human populations have acquired sequence polymorphisms in mtDNA, which are maternally inherited genetic haplogroups [47]. The association of mtDNA haplogroups with altered disease susceptibility and longevity has previously been reported [47,48,49,50,51].

During the natural ageing process, mitochondrial function declines due to accumulation of damaged mitochondrial genomes resulting from localized ROS exposure, and chronic, sterile, low-grade inflammation develops, contributing to the pathogenesis of the metabolic syndromes, obesity, and diabetes mellitus (DM). Inflammageing is the result of chronic physiological stimulation of the innate immune system [52]. Recent investigation of innate immunity has revealed insights into host reactions to noninfectious diseases and the inflammatory modulation of basic cellular metabolic processes [53]. Nutrient excess triggers inflammation in metabolic-related pathways (meta-flammation) through provoked innate immunity [54,55]. Among the meta-flammation disorders, abdominal obesity and diabetes are among the main pathologies associated with the increased burden of cellular senescence [56], and are clustered with a group of risk factors related to cardiovascular diseases [57]. As toll-like receptor 4 (TLR4) plays an important role in innate immunity, we investigated the effect of TLR4 knockout (T4KO) in a high-fat high-sucrose diet-induced obesity mouse model. In a nutrient excess condition, T4KO mice, as compared with wild type mice, had increased body weight and body fat, but decreased oxidative damage, M1/M2 macrophage ratio, chronic inflammation, and insulin resistance. This protective effect is mediated through metabolic reprogramming of mitochondria in adipose tissue. The study provides critical insight into linking innate immunity–mitochondria interplay with prevention of diabetes [58].

Insulin resistance status leads to elevated blood sugar and production of advanced glycation end products [59]. Among the advanced glycation end products, the carboxymethyl lysine glycation end products may be found in obese tissue, causing dysregulation of adipokine expression [60]. Hence, ageing could lead to a vicious cycle, which starts with immunosenescence, through the status of insulin resistance, and leads to advanced glycation end products being aggregated, and a return to inflammation status. Another phenomenon in the immune system noted in ageing is increased myeloid hematopoiesis [61]. Such increased myeloid hematopoiesis and the relatively decreased responses of adaptive immunity causes the imbalanced immunity to tend toward immunosenescence status. The production of inflammatory cytokines IL-1, IL-6, and TNF-α from the myeloid lineages accumulate. The so-called myeloid-derived suppressor cells will also increase in proportion to the increased myeloid hematopoiesis, and thus further suppressing T cell production and proliferation [61]. 

## 5. Mitochondrial DNA and Double Strand RNA Release and the Innate Immune Reaction

Mitochondria, as the cellular power plant, can function as signaling platforms to regulate cellular growth and death [62]. Mitochondria assist cells to maintain homeostasis and enable their adaptation to various stressors [63]. In addition to bioenergetic and biosynthetic functions, mitochondria are increasingly recognized as a central trigger of innate immunity that influences anti-microbial responses, autoimmune diseases, metabolic disorders, and cancers [64]. Innate pattern recognition receptors, mainly toll-like receptor 2 (TLR2) and toll-like receptor 4 (TLR4), and the nucleotide oligomerization domain (NOD) receptor protein NLRP3, trigger the activation of downstream inflammatory cascades through mitochondrial ROS [65,66,67,68]. 

Mitochondria are able to respond to cellular damage [69], through which they may accumulate oxidative stress and become irreversibly affected. Mitochondrial DNA can be released by pro-apoptotic, apoptotic, or necrotic cells, and becoming cell-free circulating mitochondrial DNA (ccf-mtDNA) [70,71]. The mtDNA can be actively secreted as ccf-mtDNA [72,73] or transported in extracellular vesicles (EVs) in response to cell stress [74]. Recent clarification of cytosolic escape of mtDNA, and that mitochondrial double strand RNA (mtdsRNA) triggers innate immunity further underscores the pivotal role of mitochondria in inflammation-related diseases [75,76]. The accumulation of mtDNA damage, as has been seen in patients with sepsis, cancer, myocardial infarction, diabetes, post-surgical intervention, and severe trauma, activates innate immune responses via different pathways, and ccf-mtDNA levels could possibly be used as a predictor of outcome in the above-mentioned diseases [77,78,79,80]. 

Collins et al. [81] were the first to report the immunostimulatory potential of mtDNA in 2004, noting that mtDNA when added to mouse splenocytes elicited TNF-α, and induced arthritis when it was released in the joints of mice. A number of other studies have shown that mtDNA can directly engage pattern-recognition receptors (PRR) of the innate immune system to enhance pro-inflammatory responses [82]. MtDNA can be recognized by three important pattern recognition receptors (PRRs) of the innate immune system, including TLR9, cytosolic inflammasomes, and type I interferon response, triggering a pro-inflammatory response [83,84]. TLR9 is a DNA-sensing PRR, and it is able to detect hypomethylated CpG motifs of DNA. MtDNA, due to its similarities with bacteria DNA, can mediate a pro-inflammatory response dependent on the TLR9 pathway [85]. In a model of shock, mtDNA was released by shock, and activated neutrophils via p38 MAPK pathway and triggered the release of MMP-8 and MMP-9 via TLR9. Furthermore, C57BL/6 mice challenged intratracheally with mtDNA induced a local inflammatory response, producing proinflammatory cytokines and activating p38 MAPK via TLR9, accompanied by macrophage infiltration [86]. Moreover, mtDNA is involved in trauma-induced neutrophil extracellular trap (NETs) formation, which is an extracellular structure secreted by neutrophils, and able to bind and kill microorganisms. MtDNA is able to induce NET formation in human neutrophils via TLR9 [87]. Plasma ccf-mtDNA activates the TLR9-mediated signaling pathway in circulating neutrophils, releasing tumor necrosis factor (TNF), interleukin-6 (IL-6), and adhesion molecules.

In addition to the extracellular mtDNA, in the intracellular space, mitochondrial damage is a critical step in inflammasome activation. There is a potential role for mtDNA serving as a direct ligand for NLRP3 and AIM2 inflammasome [88]. Zhong et al. reported that newly synthesized mtDNA and its cytosolic release play a key role in triggering NLRP3 inflammasome, a central mechanism in chronic inflammatory and degenerative diseases [76]. TLR4 engagement triggers MyD88/TRIF-dependent signaling that activates IRF1 to induce the expression of mitochondrial UMP-CMP kinase 2 (CMPK2), a mitochondrial nucleotide kinase, a rate-limiting enzyme for de novo mtDNA synthesis. This event is linked to the new synthesis of mtDNA, and this freshly generated mtDNA is thought to result in the production of oxidized mtDNA (ox-mtDNA) fragments. Subsequently, ox-mtDNA exits the mitochondria via mitochondrial pore transition (MPT), which opens after exposure to NLRP3 activators. This leads to the production of inflammatory proteins, such as interleukin one beta (IL-1β). In addition, mitochondrial apoptotic signaling causes the release of mtDNA and triggers the cytosolic DNA sensor cyclic GMP-AMP synthase (cGAS)-stimulator of interferon genes (STING) to drive interferon responses [89]. Once type I IFNs are secreted, a complex web of host defenses will be induced, with the expression of hundreds of interferon-stimulated genes [90].

Dhir et al. demonstrated mtdsRNA as a source of self-nucleic acids initiating an interferon-related innate immune response via the cytoplasmic dsRNA sensor melanoma differentiation-associated gene 5 (MDA5) and mitochondrial anti-viral signaling protein (MAVS) in the context of perturbed RNA metabolism [75]. For example, upon sensing cytoplasmic viral RNA, the retinoic-acid-inducible protein I (RIG-I)-like receptor family (RLRs), RIG-I and MDA5, interact via the outer mitochondrial membrane adaptor protein MAVS, leading to activation of downstream pathways, such as interferon regulatory factor 3 (IRF3), MAP kinases, and nuclear factor-κB (NF-κB) [91]. Clinically, free circulating mtDNA has been considered as a danger-associated molecular pattern in assessing the prognosis for myocardial infarction [92]. Our previous studies demonstrated that the disease activity of systemic lupus erythematosus (SLE) is positively associated with the mitochondrial membrane APO2.7 level of CD19+ cells, but negatively associated with MAVS and caspase-9 levels, which collectively indicate a mitochondrial pathway [93].

## 6. The Role of Mitochondria in Regeneration

Mitochondria are intracellular energy plants and signaling organelles. Importantly, current research is further elucidating the roles of extracellular mitochondria. Forms of extracellular mitochondria can be found free, enclosed by a membrane, such as inside the platelets or vesicles, or as ccf-mtDNA. When existing outside cells, they can elicit regenerative effects, or serve as an activator of immune response [94,95]. Platelets, carrying mitochondria, perform complex activities such as tissue repair [96,97,98,99]. The major pathways reported during megakaryocytes maturation are the JAK-STAT, PI3K-AKT, MAPK, and Wnt pathways, all of which are related to mitochondrial biogenesis [100]. The Wnt signaling cascade, in particular, is a key driver of tissue morphogenesis and architecture, which has been identified as a crucial factor for pluripotent capacities [101].

Mitochondria inside platelets not only produce energy but also regulate thrombus formation via equilibrium between ROS production [102,103,104]. It has been observed that the malfunction of proteins related to mitochondrial bioenergetic function has been linked to the severity of Type 2 diabetes (T2DM), sepsis, Parkinson’s disease, thrombocytopenia, and Alzheimer’s disease [105,106,107]. Overcoming multiple immune dysfunctions is fundamental for the treatment of Type 1 diabetes (T1DM), and the neogenesis of pancreatic islet β cell is a key factor in the treatment for T1DM and long-term T2DM. Previous studies have established that platelet-derived mitochondria not only exert a function in immune tolerance, but also carry the islet-specific transcription factor (MAFA), the pancreatic progenitor-associated marker (SOX9), and embryonic stem cell-related self-renewal markers (including OCT4, SOX2, KLF4, and C-MYC) [94,108]. Zhao et al. showed that mitochondria released by platelets can be immune-regulatory and have regenerative properties in diabetic patients after receiving stem cell educator therapy [109]. Platelets and their released mitochondria, after receiving stem cell educator therapy, expressed immune tolerance-related markers, and can modulate the proliferation and function of immune cells. These mitochondria also displayed embryonic stem cell- and islet β-cell-associated markers, leading to the improvement of islet β-cell functions; more specifically, there was an expression of an immune checkpoint receptor PD-1 (programmed death receptor-1) on CD4^+^ and CD8^+^ T cells after treatment with mitochondria. Additionally, immunohistochemistry data demonstrated the migration of platelets into pancreatic islets of diabetic subjects. These platelets could heal the damaged islet β-cell [109].

In recent years, mitochondrial therapy has shown promising effects in treatments for heart ischemic events, Parkinson’s disease, and fulminating hepatitis [110,111,112]. It has been proposed that mitochondrial transplantation therapy could replace damaged mitochondria, thereby leading to the repair of heart tissue [73]. In 2006, Spees et al. first demonstrated that human mesenchymal stem cells (MSCs) could transfer mitochondria through nanotubular structures to recipient cells with defective mitochondrial DNA (mtDNA) [113]. Since then, bone marrow-derived MSCs [114,115,116] and adipose-derived MSCs [115,117] have been shown to exhibit the capability of transferring mitochondria, thereby rescuing cellular bioenergetics. Recent evidence suggests that the mitochondria isolated from muscle tissue do not exacerbate the immune response when transplanted in vitro or in vivo [118]. Once inside the cell, the exogenous mitochondria, depending on their membrane constituents, can fuse with the endogenous network, with fusion-associated proteins (e.g., MFN1, MFN2, and OPA1) [72,119]. Several pathologies affect mitochondrial function and structure, including sepsis, pancreatitis, and fulminant hepatitis, thus presenting potential candidates for mitochondrial therapy [94,110,111]. 

Our colleagues have conducted a series of studies demonstrating that umbilical cord Wharton’s jelly-derived MSCs are capable of conducting mitochondrial transplantation therapy by transferring healthy mitochondria to replenish lost mtDNA [120] and reducing the mutated-mtDNA burden, thus restoring mitochondrial function in cybrid cells carrying point mutation mt.8344A > G [102], and in MELAS fibroblasts carrying point mutation mt.3243A > G. 

## 7. The Role of Mitochondria in Crosstalk between Cells

Mitochondria have been shown to be horizontally transferred between mammalian cells, with the incorporation of mitochondria into the original mitochondrial network of recipient cells, contributing to changes in the bioenergetic profile both in vitro and in vivo [121]. The transfer of mitochondria may also result in the initiation of stem cell differentiation, reprogramming cells, or activation of signaling pathways. Mitochondria exchange between cells can occur via tunneling nanotubes (TNTs), EVs, and cellular fusion [121]. TNT formation was reported in immune cells as well as in neurons, glial cells, and cardiomyocytes [122]. Recent evidence strongly supports that mitochondrial transfer does occur in vivo [74]. Astrocytes may release extracellular mitochondrial particles that enter neurons after stroke, and having a role in the neuroglial crosstalk that may contribute to endogenous neuroprotective and neural recovery mechanisms [123]. Regarding cancer, the mitochondria transfer is enhanced by some chemotherapies and confers a survival advantage to leukemic blasts and leukemia initiating cells [124]. Mitochondrial dysfunction is a key event in simulated ischemia/reperfusion (SI/R) injury. In vitro (SI/R) experiments demonstrated that bone marrow(BM)-derived mesenchymal stem cells rescue injured H9c2 cardiomyocytes, via transferring intact mitochondria through tunneling nanotubes [121]. Recent evidence has shown that in stress response, stromal cells can transfer healthy mitochondria to hematopoietic stem cells (HSC), facilitate HSC bioenergetics shift towards oxidative phosphorylation, and leukocyte expansion [125]. Mitochondria have emerged as a potential regulator of HSC fate and regenerative medicine.

## 8. mtDNA in Trauma and Organ Transplantation 

In critically ill patients, mtDNA may be released into plasma by necrosis, apoptosis, or active ejection by leukocytes. Plasma mtDNA levels are associated with ARDS in trauma or sepsis patients [80,126]. Severe trauma results in high concentrations of mtDNA release from tissues with subsequent uncontrolled inflammation and end-organ damage, and post-injury complications [127,128,129,130,131,132,133,134]. Cells that migrate to sites of injury include leukocytes, platelets, and stem cells that all these cells can release ccf-mtDNA. The process of ccf-mtDNA in response to trauma initiates from an immediate surge of ccf-mtDNA released from injured cells into the extracellular space, which through circulation results in a secondary active release of ccf-mtDNA by immune cells and platelets, followed by release of inflammatory cytokines and end-organ damage [129,135]. 

The innate immune system is a critical regulator of the adaptive immune responses that lead to allograft rejection. Mitochondrial damage and mitochondrial metabolism is critical during organ implantation and reperfusion of ischemic tissue [136,137,138,139]. Mitochondria cause ischemia-reperfusion (IR) injury by generating damaging reactive oxygen species (ROS) upon reperfusion, which lead to the release of mitochondrial damage-associated molecular patterns (DAMPs), following activation of the innate immune response, contributing to organ rejection [140,141]. Increased extracellular mtDNA is associated with elevated levels of inflammatory cytokines, and early organ dysfunction in liver and kidney transplantation [142,143,144].

Lin et al. demonstrated that extracellular mitochondria are abundant in the circulation of deceased organ donors, which correlates with early allograft dysfunction. Extracellular mitochondria led to the upregulation of endothelial cell (EC) adhesion molecules, inflammatory cytokines and chemokines, and activated dendritic cells to upregulate costimulatory molecules. They concluded that ccf-mtDNA from organ donors may directly activate allograft ECs, and result in graft rejection in transplant recipients [145].

## 9. Targeting Mitochondria as a Novel Therapeutic Strategy 

Our research indicates that a high-fat diet could lead to ROS expression, DNA and protein oxidative damage, and adipose tissue inflammation [146]; however, the inflammation could be resolved by mitochondrial manipulation, such as feeding mice with N-acetylcysteine. This phenomenon is better presented when the *N*-acetylcysteine is provided at an earlier stage, possibly due to the better flexibility of the immune reaction at a younger age, or related to the physiological effects of inflammageing [147].

As regulators of cellular energy homeostasis and cell death signaling, the continued integrity of mitochondria within a cell is crucial for its sustained health. With its discovery more than decades ago, metformin represented a milestone in the treatment of patients with T2DM. Mitochondria are the main subcellular targets of metformin, which accumulates selectively in mitochondria in many hundred-fold higher concentrations than those observed in the extracellular medium [148]. Recent evidence in humans indicated the novel pleiotropic actions of metformin, including antiproliferative, antifibrotic, and antioxidant effects [149], as well as potential anti-ageing [150], anticancer, and immune-modulatory features [151,152]. These effects are thought to occur through pathways that involve changes in the gut microbiota and both AMP-activated protein kinase-dependent and independent mechanisms [153]. 

Metformin decreased the LPS-induced production of the proinflammatory cytokine pro-IL-1β secretion in mouse bone marrow-derived macrophages, in agreement with previous findings in human monocyte-derived macrophages [154,155], and the mechanism is thought to result from the specific inhibition of mitochondrial ROS production driven by reverse electron transport at the mitochondrial respiratory chain complex 1 [154]. 

Wang et al. has demonstrated that metformin could control lupus activity via inhibiting the neutrophil extracellular trap formation and the production of anti-mitochondrial DNA autoantibodies [156]. In our in vitro transmitochondrial cybrid cell culture system, presence of the intracellular sensor of dsRNA, MDA5, could up-regulate MAVS and IRF3 in a high glucose condition with fatty acid, even without antigen-presenting cells. After the metformin was placed in the in vitro culture system, the levels of MDA5, MAVS, TBK1, and IRF3 were all significantly down-regulated [157]. Our study provided insight into the therapeutic activity of metformin in SLE patients. Further studies are required to elucidate the immune-mediated mode of action of metformin.

Glucagon-like peptide 1 receptor agonists (GLP-1RAs), initially designed to treat diabetes, have shown protective effects in several clinical trials in patients with Alzheimer’s or Parkinson’s disease. Post-treatment with GLP-1 analogues ameliorated mitochondrial dysfunction by increasing expression of Cox IV and SOD1, increasing HSP60 and PHB1, and increasing PDH expression [158]. A further study found that post-infarction treatment with GLP-1 receptor agonist, significantly reduced adverse LV remodeling and the decline of cardiac function. This paralleled a Parkin-mediated increase in autophagy, mitophagy, and mitochondrial biogenesis [159]. A previous study also showed that GLP-1RA activates the sirtuin pathway, and increases energy expenditure in human adipocytes, which may play a role in body weight reduction [160].

Although the benefits of mitochondrial transplantation therapy have been demonstrated, there are still questions surrounding the most effective methods to obtain and transfer/transplant the mitochondria. Optimizing techniques for detecting mtDNA, and mitochondria number or activity could have a positive impact in improving current clinical practices [94,161].

## Figures and Tables

**Figure 1 ijms-22-02668-f001:**
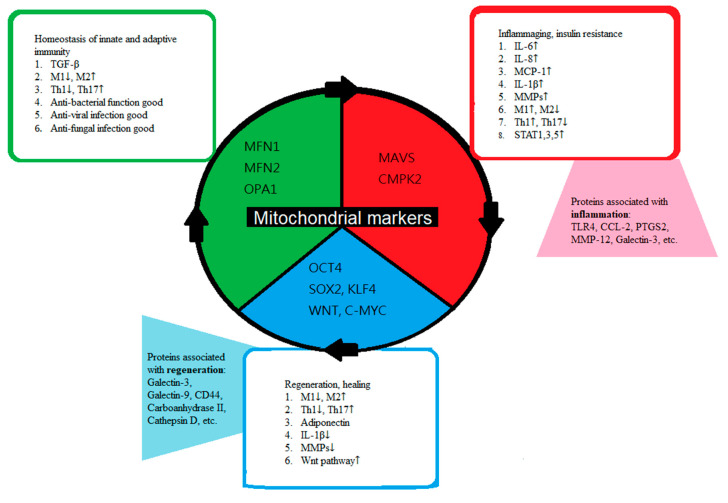
Summary of the cooperation of mitochondria (central circle) in controlling the balance between immune cells in homeostasis (green box), inflammageing (red box), and regeneration (blue box) status. Figure legends: immune cells or tissue cells in either homeostasis (green box), ageing (red box), and regeneration (blue box) status have different phenotypes and mitochondrial markers, which represent the interactions between the cell itself and its environment. The mitochondrial markers represent different mitochondrial DNA activation in each status, homeostasis (green), ageing (red), and regeneration (blue). The proteins shown inside the hexagonal boxes could be either found in regenerative status or in the inflammatory status, respectively. Abbreviations: TGF-beta, transforming growth factor beta; MFN1, mitofusin-1; MFN2, mitofusin-2; OPA1, optic atrophy-1; MAVS, mitochondrial antiviral-signaling protein; CMPK2, cytidine/uridine monophosphate kinase 2; OCT4, octamer-binding transcription factor 4; SOX2, SRY-Box transcription factor 2; KLF4, Kruppel like factor 4; C-MYC, cellular-Myelocytomatosis; MCP-1, monocyte chemoattractant protein-1; MMP, matrix metallopeptidases; STAT, signal transducer and activator of transcription; CCL-2, chemokine (C-C motif) ligand 2; PTGS2, prostaglandin-endoperoxide synthase 2.

**Table 1 ijms-22-02668-t001:** Summary of the immune cells or other cells/molecules in different states of mitochondrial condition in this review article.

		**Macrophage**	**T Cell**	**B Cell**	**Combined Immune Cells or Others**	**Mitochondrial activity**
Different phases of tissue conditions	Inflammation	Macrophage polarization in inflammatory diseases [16,17].	Down-regulation of T lymphocyte activation protects (NZB x NZW) F1 mice from lupus-like disease [29].	Plasma cell inducing clinical severity of MRL/lpr lupus-prone mice [30].	Leukocyte apoptosis, autoantibodies and disease severity in systemic lupus erythematosus [25,31,32].	
		Glycolysis [3]	Glycolysis [3]	Glycolysis [3]	Increased reactive oxygen species [3]	
	Repair/Regeneration	Macrophages: supportive cells for tissue repair and regeneration [15,20,33].	Immune mediated roles of regulatory T-cells during wound healing [21].	Metabolic reprogramming commits differentiation of human B cells [34]	Trends in tissue repair and regeneration, phagocytosis of apoptotic cells [2,22,23,24]	
		Fatty acid oxidation [3]	Fatty acid oxidation [3]	Decreased reactive oxygen species [3]	Mitochondrial fission [3]	
	Ageing	Similar epithelial phenotypes during lung development and radiation-induced fibrogenesis [20].	Type 2 diabetes is associated with the accumulation of senescent T cells [35,36].	Regulation of energy metabolism during early B lymphocyte development [37,38]	Macrophages with a potential beneficial role in atherosclerosis [26,39,40].	

## Data Availability

Not applicable.

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
