# Peer review of "The Role of Mitochondria in Immune-Cell-Mediated Tissue Regeneration and Ageing"

_ijms, 2021, doi:10.3390/ijms22052668_

Round 1
Reviewer 1 Report
The authors have sufficiently addressed my comments and the article is good for publication in its current form.
Reviewer 2 Report
The paper has a number of significant improvements and even some additional topics, like the information about free circulating mitochondrial DNA. To me, it looks quite good and it is very interesting to read. I have no complaints, so I would just publish it. There are some polymorphisms in TLR4 receptors, which make it somewhat hypo-functional, but they can make some patients even resistant to the development of Alzheimer's disease. Although it seems these people can be more susceptible to inflectional diseases. But this is not the topic of this paper.Reviewer 3 Report
The authors have satisfactorily addressed the comments and the paper is suitable for publication.
This manuscript is a resubmission of an earlier submission. The following is a list of the peer review reports and author responses from that submission.
Round 1
Reviewer 1 Report
A review article is a text that contains a summary of the more recent research on certain topics. The present Review appears not focused on the topic declared in the title. The first section after the Introduction (Immune cells under....) is too generic and mitochondria are not mentioned at all. The second section does not elaborate on the role of mitochondria in aging forgetting to cite important studies (PMID: 24470107). The third section on tissue regeneration could be removed as well as the section on haplogroups. Also, the section on (free) mitochondrial DNA should be explained in much since there a lot of studies on free circulating mtDNA in different physio-pathological conditions. Moreover, Table 1 does not add any useful information to the topic of the paper and could be removed. Essentially, the paper must be profoundly revised and made more focused, mainly following Figure 1. Authors have to broaden and deepen the role of mitochondria in inflammaging and regeneration avoiding generic introduction.
Reviewer 2 Report
Thank you for the paper. It was interesting to read. In general, it looks good. However, on the page 15, where authors discuss the action of metformin in mitochondria they suddenly insert two sentences about the role of GLP-1. Native GLP-1 peptide has a half-life of few minutes because it is inactivated by DPP4 and it is very short. Therefore, the analogues were generated (Exendin-4, liraglutide, dulaglutide). Exendin-4 can generate the immune response, because it is originally from the lizard. Dulaglutide seems to be the best, because it is GLP-1-Fc-hybrid and it has the longest half-life. Anyway, metformin and GLP-1 work with completely different mechanisms. In general, the biguanides and are particularly known for their inhibiting effect on glucose synthesis by blocking the FBPase. GLP-1 analogues stimulate insulin secretion by beta cells and inhibit glucagon secretion in alpha cells. I would add some more sentences illustrating this contrast also in terms how particularly it could work in mitochondria.
Minor issues:
Page 8: last sentence “SLE-prone (NZB × NZW) F1 mice”. SLE means “Systemic lupus erythematosus”?
Page 9: “induced plasma cell apoptosis. [34, 35]_ENREF_21 Furthermore, galectin-3 has”. Is that a formatting mistake?
Reviewer 3 Report
Thank you for the opportunity to review your manuscript. The review article is on a very fascinating topic that recently has become increasingly highlighted in innate immune and tissue repair literature. Unfortunately, the manuscript is not well-organized. The first half is a review of basic immunology. The second lists recent findings in the field and sites recent papers, but doesn't add novel discussion. The two halves need to be better blended into a coherent manuscript that focuses on the specific subject mentioned in the title, namely the role that mitochondria play in inflammation, disease, and tissue repair. In its current state, it reads like a rambling summary of various tidbits of information. Also, the review has skipped over several recent key papers on extracellular mitochondria in the setting of trauma and organ transplantation. Several errors in grammar and word use need to be corrected (e.g. fulminant hepatitis, not fulminating hepatitis, etc).